# Downregulation of the Tumor Suppressor TFF1 Is Required during Induction of Colon Cancer Progression by L1

**DOI:** 10.3390/cancers14184478

**Published:** 2022-09-15

**Authors:** Arka Saha, Nancy Gavert, Thomas Brabletz, Avri Ben-Ze’ev

**Affiliations:** 1Department of Molecular Cell Biology, Weizmann Institute of Science, Rehovot 7610001, Israel; 2Department of Experimental Medicine I, Nikolaus-Feibiger-Center for Molecular Medicine, University of Erlangen-Nuernberg, 91054 Erlangen, Germany

**Keywords:** TFF1, L1, cancer cell adhesion and invasion, colon cancer, metastasis

## Abstract

**Simple Summary:**

Aberrant activation of Wnt/β-catenin signaling and the subsequent induction of downstream target genes is a hallmark of colorectal cancer (CRC) development. Previously, we found that overexpression of the immunoglobulin-like cell adhesion receptor L1CAM (L1), a target of the Wnt/β-catenin pathway, confers enhanced proliferation, motility, tumorigenesis, and liver metastasis in CRC cells. Transcriptomic and proteomic analyses revealed changes in both pro-tumorigenic and potential tumor-suppressor genes in L1-overexpressing CRC cells. We wished to identify such tumor suppressor/s, and found that trefoil family factor 1 (TFF1) was involved in L1-mediated CRC progression. TFF1 overexpression suppressed the growth, motility and tumorigenesis of L1-expressing CRC cells by inhibiting the NF-κB pathway. In human CRC tissue, TFF1-positive staining was evident in goblet cells of the normal mucosa, while in CRC tissue, TFF1 expression was lost in >50% of the tumor samples. Our results support a tumor-suppressor role of TFF1 in human CRC, and we suggest that TFF1 could be used for CRC detection and as a novel therapeutic target in L1-mediated CRC.

**Abstract:**

The immunoglobulin family cell adhesion receptor L1 is induced in CRC cells at the invasive front of the tumor tissue, and confers enhanced proliferation, motility, tumorigenesis, and liver metastasis. To identify putative tumor suppressors whose expression is downregulated in L1-expressing CRC cells, we blocked the L1–ezrin–NF-κB signaling pathway and searched for genes induced under these conditions. We found that TFF1, a protein involved in protecting the mucus epithelial layer of the colon, is downregulated in L1-expressing cells and displays characteristics of a tumor suppressor. Overexpression of TFF1 in L1-transfected human CRC cells blocks the pro-tumorigenic and metastatic properties conferred by L1 by suppressing NF-κB signaling. Immunohistochemical analyses revealed that human CRC tissue samples often lose the expression of TFF1, while the normal mucosa displays TFF1 in goblet cells. Identifying TFF1 as a tumor suppressor in CRC cells could provide a novel marker for L1-mediated CRC development and a potential target for therapy.

## 1. Introduction

Hyperactivation of the Wnt/β-catenin signaling pathway and the subsequent induction of β-catenin/TCF target genes is a hallmark of CRC development [1]. We previously identified members of the immunoglobulin-like family of cell adhesion receptors NrCAM and L1 as targets of the Wnt/β-catenin pathway in CRC cells [2,3]. We reported that L1 is localized exclusively at the invading front of human CRC tissue [3]. Overexpression of L1 in CRC cells results in enhanced proliferation, motility and tumorigenesis, and promotes the metastasis of CRC cells to the liver in a mouse model [3,4]. A key downstream signaling mechanism involved in L1-mediated tumorigenesis is an L1–ezrin–NF-κB pathway that includes the activation of several genes which, when expressed in CRC cells, can mimic the effects conferred by L1 overexpression [5,6]. The identification of the pro-tumorigenic properties of such genes by transcriptomic, proteomic, and secretomic analyses demonstrated their increased levels after L1 expression in CRC cells [1,6,7,8,9,10,11,12,13,14]. Since tumor progression includes, in addition to the induction of pro-tumorigenic genes, the downregulation of tumor suppressor genes, we wished to investigate the possibility that the L1-mediated CRC progression also involves the inhibition of tumor suppressors. Toward this aim, we conducted a transcriptomic analysis of genes whose expression increases when the L1–ezrin–NF-κB pathway is inhibited [6]. By adopting this approach, we identified trefoil family factor 1 (TFF1) as a potential tumor suppressor in L1-mediated CRC progression. TFF1 belongs to a highly conserved group of small proteins that are expressed and secreted by mucus-secreting cells, and plays an essential role in protecting the mucus layer of the stomach and intestine [15]. The involvement of TFF1 in tumor progression has been reported for various types of cancer [16,17,18,19], but its role in L1-dependent CRC development has not been determined. In the present study, we investigated the possible role of TFF1 in L1-mediated CRC progression.

## 2. Materials and Methods

### 2.1. Cell Culture

The cell lines LS 174T (ATCC CL-188) and DLD-1 (ATCC CCL-221) were obtained from ATCC and cultured in RPMI-1640 medium (Gibco, Thermo Fisher Scientific, Paisley, UK) containing 10% FBS (Gibco, Thermo Fisher Scientific, Paisley, UK), supplemented with 1% penicillin/streptomycin solution (Biological Industries, Beit-HaEmek, Israel). LS 174T-L1 and DLD-1-L1 cells were grown in RPMI-1640 medium containing neomycin (800 µg/mL). LS 174T-L1 + TFF1, LS 174T-L1 + shp65 and LS 174T-L1 + IκB-SR cells were grown in RPMI 1640 medium supplemented with both neomycin (800 µg/mL) and puromycin (10 µg/mL).

### 2.2. Plasmids

The TFF1 cDNA expression vector (pQXCIP) was obtained from Dr. Alessandra Tosco (Department of Pharmacy, Biomedical Division “Arturo Leone”, University of Salerno, Salerno, Italy).

### 2.3. Transfection, Cell Proliferation, and Motility Assays

The TFF1 cDNA expression vector was transfected into LS 174T-L1 cells using the Xfect™ transfection reagent (TaKaRa, Mountain View, CA, USA) as suggested by the manufacturer. Cell proliferation under stress was performed by seeding 5000 cells in 12-well plates containing RPMI-1640 medium supplemented with 0.5% FBS. The rate of cellular proliferation was determined by cell counting each day for five days.

Cell motility was determined by the “scratch wound closure” assay in triplicate samples as described in [6]. A total of 10^5^ cells were seeded in 12-well plates and grown to form a confluent monolayer. Artificial wounds were introduced into the monolayers using a micropipette. Fresh medium containing 5 μg/mL Mitomycin C was added to inhibit cell proliferation. Pictures were taken at 0 h and 24 h after introducing the wound in triplicate samples for each cell clone. The percent wound closure was determined by FIJI software (v.1.53c, NIH, Bethesda, MD, USA) and plotted by GraphPad Prism 8 software (San Diego, CA, USA, www.graphpad.com).

### 2.4. Western Blotting and Immunofluorescence

Western blotting was carried out with cell lysates prepared in RIPA lysis buffer containing a 1% protease inhibitor cocktail as described in [7]. Forty micrograms of protein per lane were resolved by 10% SDS-PAGE, transferred onto nitrocellulose membranes, and probed with the following antibodies: rabbit anti-L1 at 1:2000 dilution (provided by Dr. V. Lemmon, University of Miami, Miami, FL, USA), mouse anti-TFF1, sc-271464 at 1:500 dilution (Santa Cruz Biotechnology Inc., Dallas, TX, USA), rabbit anti-phospho-IκB #2859 at 1:1000 dilution (Cell Signaling Technologies Inc., Danvers, MA, USA), rabbit anti-NF-κB p65, sc-109, at 1:1000 dilution (Santa Cruz Biotechnology, Inc., Dallas, TX, USA), and mouse anti-tubulin at 1:5000 dilution (Sigma-Aldrich, St. Louis, MO, USA). Western blots were developed using the ECL method (Amersham Biosciences, Buckinghamshire, UK), and the bands were visualized using the ChemiDoc MP imaging system (Bio-Rad Laboratories, Haifa, Israel).

Cells grown on coverslips were used for the immunofluorescence assays following cell permeabilization with 0.5% Triton X-100 and fixation with 4% paraformaldehyde. The fixed cells were blocked with 5% horse serum in PBS, and probed with rabbit anti-L1 (1:200 dilution) and mouse anti-TFF1 antibodies (1:100 dilution). Goat anti-mouse IgG labeled with Alexa Flour-488 (ABCAM, Trumpington, Cambridge, UK) and goat anti-rabbit IgG labeled with Cy3 (Jackson Immunoresearch Laboratories, West Grove, PA, USA) secondary antibodies were used at a 1:1000 dilution. Nuclei were stained with 5 µg/mL 4′-6-diamidino-2-phenylindole (DAPI, Sigma-Aldrich, St. Louis, MO, USA). Images were captured by the Zeiss LSM 800 confocal microscope and ZEN imaging software (Carl Zeiss Microscopy GmbH, Jena, Germany), and analyzed using ImageJ and FIJI software.

### 2.5. Dot Blotting

Dot blotting was used to detect the quantities of TFF1 protein in cells. Cell lysates were prepared using the RIPA lysis buffer supplemented with a 1% protease inhibitor cocktail. The amount of protein was quantified using a Pierce BCA protein assay (Thermo Fisher Scientific, Paisley, UK), in accordance with the manufacturer’s protocol. Serial dilutions of cell lysates between 40 µg and 2.5 µg were loaded onto a 0.45 µm pore diameter nitrocellulose membrane (Bio-Rad Laboratories, Haifa, Israel). The membranes were blocked with 5% BSA in PBS supplemented with 0.5% Tween-20 (Bio-Basic Canada, Inc., Markham, ON, Canada). The membranes were probed with a mouse anti-TFF1 antibody, sc-271464 (Santa Cruz Biotechnology Inc., Dallas, TX, USA) at a 1:250 dilution, and incubated at room temperature for 2 h followed by incubation for 2 h with a goat anti-mouse HRP-conjugated secondary antibody (ABCAM, Trumpington, Cambridge, UK), diluted 1:1000. The blots were developed using the ECL method (Amersham Biosciences, Buckinghamshire, UK), and the signal strength was determined with the ChemiDoc MP imaging system (Bio-Rad Laboratories, Haifa, Israel). Each dot blot was quantified using ImageJ and FIJI software, and the integrated densities were plotted using GraphPad Prism 8 software.

### 2.6. Quantitative RT-PCR

RNA isolation from cells was performed using the Bio-Tri reagent (Bio-Lab, Jerusalem, Israel), in accordance with the manufacturer’s protocol. First-strand cDNA synthesis from the isolated RNA was carried out using a SuperScript™ II Reverse Transcriptase kit (Thermo Fisher Scientific, Waltham, MA, USA), in the manner suggested by the manufacturer [13]. Appendix A describes the primers used for the amplification of target genes. Gene expression was calculated using the ΔΔCT method with QuantStudio Design and Analysis software v1.5.1 (Thermo Fisher Scientific, Waltham, MA, USA) and plotted as fold change in RNA levels using GraphPad Prism 8 software.

### 2.7. Tumor Growth and Metastasis Assays

Five-week-old male athymic nude mice (Foxn-1u) were used to determine subcutaneous tumor growth, as described in [4,13]. In summary, 3 × 10^6^ cells suspended in 100 µL of PBS were injected at different sites into the flanks of the mice [5]. Tumor growth was determined 14 days after injection. The tumors were excised, weighed, and photographed, and graphs were plotted showing tumor weight using GraphPad Prism 8 software.

Metastasis was determined by examining the ability of CRC cells to migrate from the spleen to the liver in athymic male nude mice after injecting 3 × 10^6^ cells suspended in 20 µL of PBS into the distal tip of the spleen, as described in [4,13]. To anesthetize the mice, Xylazine and Ketamine were injected into the peritoneum during the process. The mice were sacrificed after six weeks, and the presence of tumors formed at the site of injection in the spleen, and of liver metastases, was determined.

### 2.8. Ethics Approval

The Weizmann Institutional Animal Care and Use (IACUC) ethics committee reviewed, approved, and supervised the animal studies.

### 2.9. Immunohistochemistry

Immunohistochemical analysis was performed on 3 μm-thick tumor tissue sections from 38 paraffin-embedded human colorectal adenocarcinoma cases and on the adjacent normal tissue with the rabbit anti-TFF1 antibody (Sigma HPA003425) at a dilution of 1:2500, as previously described [3].

### 2.10. DNA Microarrays

RNA was extracted from two individually isolated CRC cell clones of LS 174T cells expressing L1, and compared with RNA extracted from two individually isolated clones of LS 174T cells expressing L1 + IκB-SR, L1 + shEzrin, or the empty vector. The cDNA microarray analysis of the extracted RNA was performed using the Affymetrix 1.0 ST GeneChips at the Weizmann Institute Microarray Facility. The chips were processed, and the data were analyzed as described in [20].

### 2.11. Statistical Analysis

GraphPad Prism 8 software was used for statistical analysis, and the significance of the data was calculated using Student’s unpaired *t*-test. Results with a *p*-value < 0.05 were considered statistically significant and were represented on the graphs with an asterisk.

## 3. Results

### 3.1. Downregulation of TFF1 in Human CRC Cells Overexpressing L1

To investigate the possibility that there is a downregulation of tumor suppressors during L1-mediated CRC development, the signaling by the L1–ezrin–NF-κB pathway was inhibited in L1-overexpressing cells, using an shRNA against ezrin [6]. Among the genes whose levels were upregulated under these conditions [6], we identified TFF1, which was reported to act as a tumor suppressor in some studies [19,21,22]. The results summarized in Figure 1A,B demonstrate that TFF1 RNA levels were reduced in both LS 174T CRC cells (expressing a mutant β-catenin) and in DLD-1 CRC cells (expressing a mutant APC) when L1 levels were increased in these cells. A decrease in TFF1 protein was observed (Figure 1C–E) in parallel with the reduction in the TFF1 RNA level (Figure 1A) in both CRC cell types, demonstrating that TFF1 levels are reduced by L1 in human CRC cells where different mutations in the Wnt/β-catenin pathway activate β-catenin/TCF signaling.

### 3.2. Overexpression of TFF1 in L1-Expressing CRC Cells Suppresses Their Growth, Motility, and Tumorigenesis

We wished to explore the possibility that TFF1 acts as a tumor suppressor during L1-mediated CRC progression. For this, we isolated CRC cell clones that overexpress both TFF1 and L1 (Figure 2A,B). The quantitative Western dot blots shown in Figure 2A,B demonstrate that in these CRC cell clones (L1 + TFF1 cl1 and cl2), the levels of TFF1 were restored, and were even higher than those observed in the control LS 174T cells expressing an empty vector (pcDNA3) (Figure 2B).

The intensity of immunostaining for TFF1 was visible in the pcDNA3-transfected CRC control cells (Figure 2C, pcDNA3), mainly in the secretory granules, as expected from the role of TFF1 as a secreted protein. In L1-expressing CRC cells, the intensity of TFF1 immunostaining was greatly reduced (Figure 2C, L1), while in cells overexpressing both L1 and TFF1, the strength of TFF1 staining was increased (Figure 2C, L1 + TFF1 cl1, and cl2).

We analyzed the growth, motility, and tumorigenic properties of LS 174T CRC cell clones overexpressing both L1 and TFF1, as compared with the control pcDNA3 transfected cells and cells overexpressing L1 (Figure 3). The results summarized in Figure 3A demonstrate that the increased proliferation of CRC cells under stress in low serum (0.5%), conferred by L1, was suppressed in L1-expressing CRC cell clones transfected with TFF1 (Figure 3A, L1 versus L1 + TFF1 cl1 and cl2). The increased motile capacity induced by L1 in CRC cells, as determined by the “scratch wound” closure method, was also suppressed upon the increase in TFF1 levels (Figure 3B, L1 + TFF1 cl1 and cl2 compared with L1, and Appendix A). The tumorigenic capacity of L1-expressing CRC cells in mice injected subcutaneously with CRC cells was also inhibited upon increasing the level of TFF1 in L1-expressing CRC cells (Figure 3C,D). Finally, the ability to form liver metastases in mice injected with these cell clones in the spleen was also reduced when the level of TFF1 was increased (Figure 3E,F). Taken together, these results suggest that TFF1 acts as a tumor suppressor during L1-mediated tumorigenesis and that its downregulation in CRC cells is a necessary step for tumor progression conferred by L1.

### 3.3. Opposing Effects of L1 and TFF1 on NF-κB Signaling during CRC Development

Since L1-mediated signaling in CRC cells involves the NF-κB pathway [5,6], we wished to examine the possibility of an interplay between TFF1 and NF-κB signaling during CRC progression conferred by L1. The results shown in Figure 4A demonstrate that in L1-overexpressing CRC cells, there was an increase in the expression of components of the NF-κB pathway, the p65 subunit of NF-κB and IκB (see Figure 4A, which compares L1-expressing cells with control pcDNA3 empty-plasmid-transfected cells). This increase in the level of NF-κB pathway components conferred by L1 was blocked when TFF1 was transfected into these cells (Figure 4A, L1 + TFF1 cl1, and cl2). The inverse correlation between TFF1 expression and NF-κB activity in L1-expressing cells was also demonstrated when NF-κB signaling was inhibited in L1-transfected CRC cells with an shRNA against the p65 NF-κB subunit (L1 + shp65) or by using the mutant IκB super-repressor (L1 + IκB-SR). Under these conditions, TFF1 RNA (Figure 4B) and protein levels (Figure 4C,D) were restored to those of the pcDNA3-expressing CRC control cells. Finally, a microarray analysis of genes upregulated in L1-overexpressing cells in which NF-κB signaling was inhibited by IκB-SR expression also included the TFF1 gene (Appendix A). Together, these results show that the NF-κB pathway and TFF1 expression are inversely related: an increase in TFF1 levels suppresses the expression of NF-κB pathway components and, conversely, suppression of NF-κB signaling restores/elevates the expression of TFF1 RNA (Figure 4B, Appendix A) and protein levels (Figure 4C,D) in L1-expressing CRC cells.

### 3.4. Localization of TFF1 in Human CRC Tissue and in Normal Mucosa

We determined the localization of TFF1 in 38 CRC tissue samples embedded in paraffin and immune-stained for TFF1 with anti-TFF1 antibody (Figure 5). Normal mucosa was positive for TFF1 in goblet cells (Figure 5A(N),F, arrows), indicative of a positive role of TFF1 in determining the differentiated phenotype of these cells. The adjacent cancer tissue lost the expression of TFF1 in 50% of the samples (19/38) (Figure 5A(C),C). In 24% of cases (9/38), the tumor tissue displayed a mixed staining pattern, with partial loss of TFF1 staining and some positive areas for TFF1 (Figure 5D). In 36% of cases (10/38), the positive tumor displayed a mucinous differentiated phenotype, which could indicate that these tumors were derived from goblet cells (Figure 5E). A summary of the different distribution of TFF1 localization in CRC tissue of the 38 CRC cases is shown in Appendix A.

Overall, these results showed that while the normal colonic mucosa displayed the presence of TFF1 in goblet cells, in 50% of the samples the adjacent tumor tissue had lost the expression of TFF1. When the tumor was positive for TFF1 it was often associated with a mucinous phenotype, indicating that such tumors originated from goblet cells of the mucosa.

## 4. Discussion

In the present study, we have shown that the progression of CRC development conferred by L1 expression includes a downregulation in the level of the tumor suppressor TFF1. Interfering with signaling by the L1–ezrin–NF-κB pathway using a shEzrin construct, we detected TFF1 among the genes whose expression was upregulated when L1 signaling via ezrin was blocked. The potential of TFF1 to act as a tumor suppressor in L1-mediated CRC development is supported by our findings, which suggest that increasing TFF1 expression in L1-transfected CRC cells results in the inhibition of CRC cell proliferation, motility, tumorigenesis, and liver metastasis. Moreover, in human CRC tissue samples, we detected a loss of TFF1 in the cancer tissue, while in the adjacent normal mucosa TFF1 was detected in goblet cells of the colon (Figure 5; Appendix A). Our results strongly support the notion that TFF1 acts as a tumor suppressor in L1-mediated CRC progression.

It has been suggested that TFF1 and other TFF family members play an essential role in mucosal protection and repair [15,23], and the involvement of TFF1 downregulation in tumorigenesis was demonstrated in a variety of cancer cells, including gastric carcinoma [17,24,25], breast cancer [26], hepatocellular cancer [27], and pancreatic cancer [28]. Studies in CRC tissue have detected both a pro-tumorigenic effect [29] and tumor-suppressive effects of TFF1 in CRC [30,31].

Several mechanisms for TFF1 action have been suggested, such as resistance to apoptosis [32], the promotion of cell migration [33,34], and the inhibition of epithelial–mesenchymal transition [28]. While many potential interacting partners for TFF1 have been reported [23], the molecular mechanisms underlying TFF1 action remain elusive. Our findings support an interplay between TFF1 and L1-mediated NF-κB signaling and demonstrate that NF-κB signaling inhibits TFF1 expression. In contrast, increased TFF1 expression suppresses the levels of critical components in the NF-κB pathway (Figure 4), therefore suggesting that TFF1 is a negative regulator of NF-κB signaling. These results support our previous study demonstrating that L1 promotes a transcriptional inhibition of c-KIT expression by NF-κB [20] in CRC cells. Our results also agree with a study that demonstrated a progressive decrease in TFF1 and an increase in NF-κB activation during gastric cancer development [24]. Moreover, in a DNA-microarray analysis of genes whose expression was upregulated when the NF-κB pathway was inhibited in L1-overexpressing CRC cells, we found TFF1 among the upregulated genes (Appendix A).

Overall, this study points to TFF1 as a tumor suppressor during L1-mediated CRC development, and provides an additional target for diagnosis and CRC therapy.

## 5. Conclusions

In the current study, we investigated the role of TFF1 activity in L1-mediated colon cancer development and progression. We report that TFF1 transfection into L1-expressing CRC cells inhibits cell proliferation, motility, tumorigenesis, and liver metastasis, and that TFF1 works as a negative regulator of NF-κB signaling by inhibiting the levels of critical components in the NF-κB pathway. We further observed a loss of TFF1 in cancer tissues and positive TFF1 signals in goblet cells of the normal mucosa of human CRC samples. The results outlined the role of TFF1 as a tumor suppressor in L1-mediated CRC development—a role which can be developed as a potential therapeutic target for CRC.

## Figures and Tables

**Figure 1 cancers-14-04478-f001:**
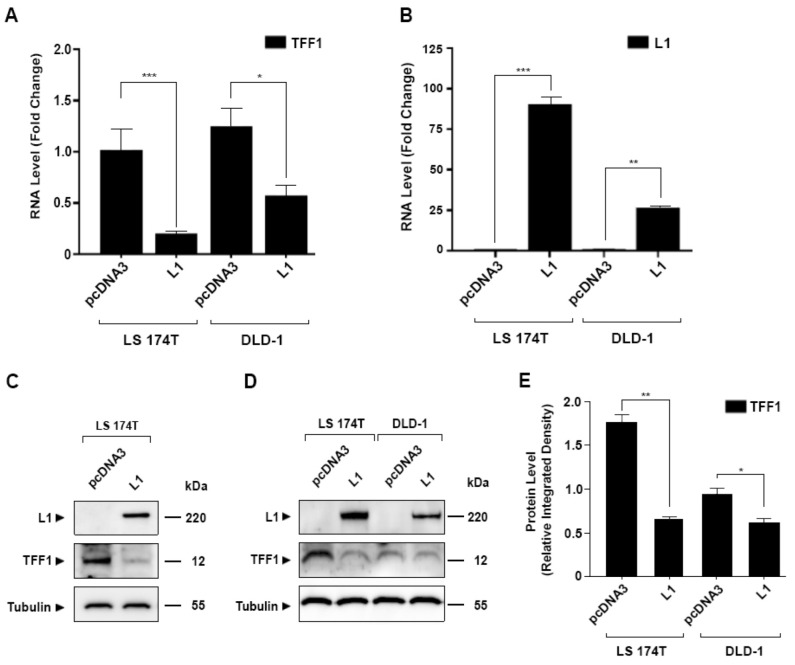
Suppression of TFF1 expression in CRC cells overexpressing L1. (**A**) The level of TFF1 RNA was determined in L1 overexpressing LS 174T and DLD-1 CRC cells; (**B**) L1 RNA levels were determined by RT-PCR in the cells described in (**A**); (**C**,**D**) the expression of TFF1 and L1 protein was determined by Western blotting in the cell lines described in (**A**). Tubulin served as loading control; (**E**) the intensity of the TFF1 bands shown in the Western blots of the different CRC cell lines (**D**) was determined by densitometric analysis of the gels shown in (**D**). Relative integrated band densities of TFF1 were calculated from triplicate gels, and the significance was determined using Student’s unpaired *t*-test. * *p* < 0.05, ** *p* < 0.01, *** *p* < 0.001. The uncropped blots are shown in Appendix A.

**Figure 2 cancers-14-04478-f002:**
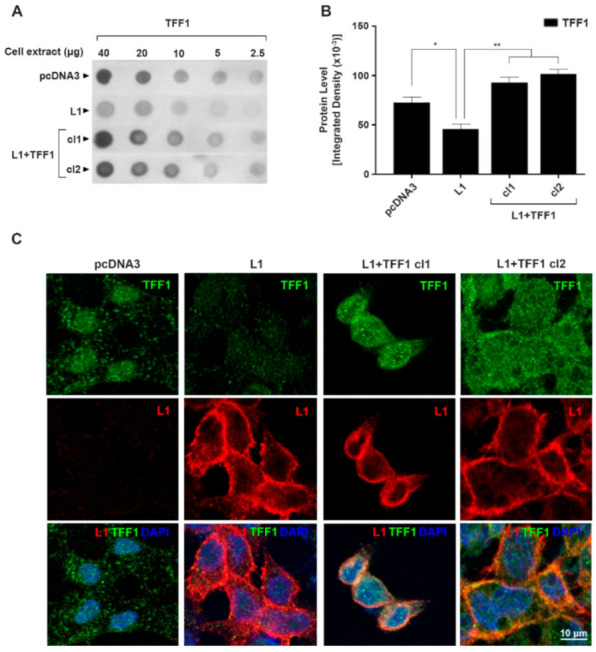
Isolation and localization of L1 and TFF1 in CRC cell clones overexpressing both L1 and TFF1. (**A**) Serial dilutions of cell extracts analyzed by immuno-dot-blotting using LS 174T CRC cell clones overexpressing both L1 (L1) and TFF1 (L1 + TFF1 cl1 and cl2); (**B**) densitometric analysis of the dot blots shown in (**A**); (**C**) localization of L1 and TFF1 in CRC cell clones overexpressing both L1 (red) and TFF1 (green) by double immunofluorescence microscopy. Nuclei were stained with DAPI (blue). * *p* < 0.05, ** *p* < 0.01. The uncropped blots are shown in Appendix A.

**Figure 3 cancers-14-04478-f003:**
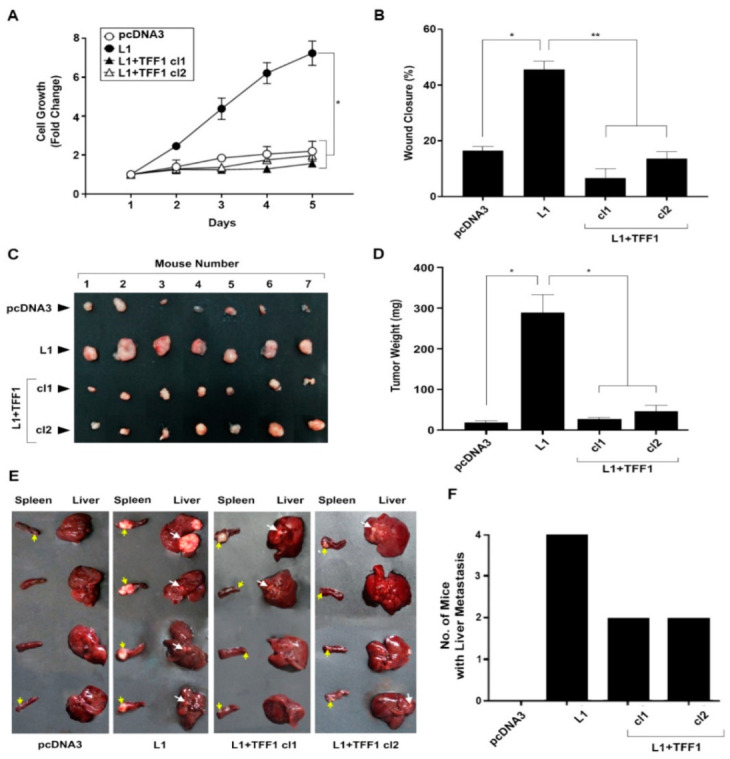
Suppression of the L1-mediated increase in cell proliferation, motility, tumorigenicity, and metastasis by the expression of TFF1. (**A**) The proliferation of pcDNA3-transfected control LS 174T CRC cells (pcDNA3), L1-overexpressing CRC cells (L1), and L1 + TFF1-overexpressing CRC cell clones (L1 + TFF1 cl1 and cl2) in the presence of 0.5% serum was determined over five days; (**B**) the motility of the CRC cell clones described in (**A**) was determined by the “scratch wound” closure method 24 h after introducing the “wound”, as percent wound closure; (**C**,**D**) the tumorigenic ability of the CRC cell clones described in (**A**) was determined by subcutaneous injection of the CRC cell clones described in (**A**); two weeks after injection, the tumors were excised, photographed (**C**), and their weight determined (**D**); (**E**,**F**) the metastatic capacity of the cell lines described in (**A**) was determined by injecting each cell clone into groups of 4 mice and determining the development of liver metastases after six weeks. The yellow arrows point to tumor growth at the site of injection in the spleen, while the white arrows point to metastatic growth in the liver of the corresponding mice. * *p* < 0.05, ** *p* < 0.01.

**Figure 4 cancers-14-04478-f004:**
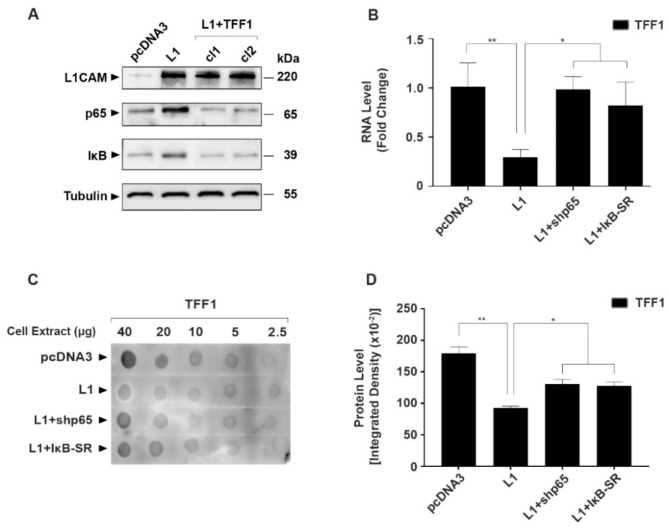
The inverse relationship between the expression of TFF1 and NF-κB signaling components in L1-expressing CRC cells. (**A**) The levels of p65 and IκB protein were determined in the CRC cell clones described in Figure 3; (**B**) the level of TFF1 RNA was determined in L1-expressing CRC cell clones in which NF-κB signaling was blocked by either an shRNA to p65 (L1 + shp65) or by the expression of the IκB super-repressor (L1 + IκB-SR); (**C**,**D**) quantitative immuno-dot-blot analysis of TFF1 protein levels in the CRC cell clones described in (**B**). * *p* < 0.05, ** *p* < 0.01. The uncropped blots are shown in Appendix A.

**Figure 5 cancers-14-04478-f005:**
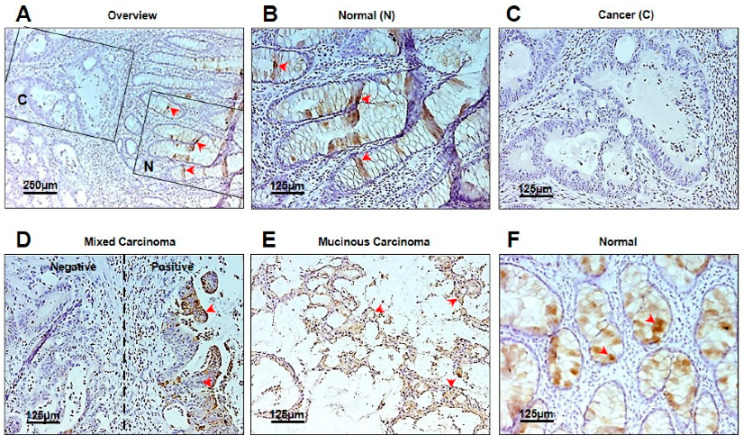
Immunohistochemical analysis of TFF1 expression in normal mucosa and in CRC tissue. (**A**) Overview of an area containing normal mucosa (N) and the adjacent cancer tissue (C). Arrows mark TFF1-positive goblet cells; (**B**) localization of TFF1 in goblet cells of normal mucosa (N) (arrows); (**C**) loss of TFF1 expression in adjacent cancer (**C**) tissue; (**D**) a mixed pattern of carcinoma tissue staining with both positive (arrows) and negative areas; (**E**) a mucinous carcinoma area positive for TFF1 staining (arrows); (**F**) normal mucosa stained for TFF1 in goblet cells (arrows).

## Data Availability

The data presented in this study are available on request from the corresponding author.

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
