# Peer review of "Downregulation of the Tumor Suppressor TFF1 Is Required during Induction of Colon Cancer Progression by L1"

_cancers, 2022, doi:10.3390/cancers14184478_

Round 1
Reviewer 1 Report
This complete study by a highly respected scientist in the field adds further understanding to the role of TFF down regulation in cancer by analyzing colon cancer. The role of NFKB as a transcriptional intermediary addresses a mechanistic missing piece in the field. The clinical translation to early detection or therapeutics in L1 mediated CRC is of novel interest. The studies are conducted carefully with appropriate controls, clear and statistically significant results.
Author Response
This complete study by a highly respected scientist in the field adds further understanding to the role of TFF down regulation in cancer by analyzing colon cancer. The role of NFKB as a transcriptional intermediary addresses a mechanistic missing piece in the field. The clinical translation to early detection or therapeutics in L1 mediated CRC is of novel interest. The studies are conducted carefully with appropriate controls, clear and statistically significant results
Response:
We thank the reviewer for their positive remarks regarding the publication of our manuscript.
Reviewer 2 Report
This manuscript is interesting and clear. However, I would like to make some considerations.
Experimental design is appropriate.
It is necessary to adjust to the numbering of the additional tables. I would propose to indicate the table of primers used in qRT as Table S1.
MW markers are missing for gels containing L1 and tubulin.
The quality of the images regarding the dot blot is not good, especially at L1.
Photos of wound scratch are missing.
The Authors identify the putative tumor suppressor gene TFF1 whose expression is downregulated in L1-expressing CRC cells. The discovery of the oncosuppressive activity of TFF1 was carried out through the blocking of the L1-ezrin-NF-κB signaling pathway. According to the Authors, it could open interesting scenarios both in the identification of markers involved in the development of CRC and in new therapeutic options.
In 2019, Yusufu et al. in “TFF3 and TFF1 expression levels are elevated in colorectal cancer and promote the malignant behavior of colon cancer by activating the EMT process” suggested TFF1 as CRC biomarker in other cell lines and detected the increased expression levels of both TFF1 and 3 in tumor tissues compared with adjacent normal ones. This reference is not mentioned in the article and could instead be the subject of discussion.
Author Response
Point 1: It is necessary to adjust the numbering of the additional tables. I would propose to indicate the table of primers used in qRT as Table S1.
Response 1: We have changed the numbering (order) of the Supplementary Tables, as suggested by the reviewer.
Point 2: MW markers are missing for gels containing L1 and tubulin
Response 2: The original (raw) western blots with the pre-stained MW markers and the segments of the blots used for each immunoblot can be seen in the “raw western blots” file.
Point 3: The quality of the images regarding the dot blot is not good, especially at L1.
Response 3: We have replaced the L1 dot blot with a clearer image. Our densitometric analysis of the L1 dot blots indicates statistical significance in L1 level changes on the dot blots, as shown in the Figures.
Point 4: Photos of wound scratch are missing.
Response 4: In the new Supplementary Fig.1, we have included photos of the wound healing experiment.
Point 5: In 2019, Yusufu et al. in “TFF3 and TFF1 expression levels are elevated in colorectal cancer and promote the malignant behavior of colon cancer by activating the EMT process” suggested TFF1 as CRC biomarker in other cell lines and detected the increased expression levels of both TFF1 and 3 in tumor tissues compared with adjacent normal ones. This reference is not mentioned in the article and could instead be the subject of discussion.
Response 5: We have included and discussed the reference by Yusufu et al. suggesting that TFF1 is elevated in CRC tissue (Ref #29, line 499). We have also included two additional references (Ref #30 and #31) that reported the opposite, namely, that TFF1 suppresses the growth and tumorigenesis of CRC cells and tissue (lines 499-500).
Round 2
Reviewer 2 Report
The manuscript can be accepted with the changes made.